# Effects of *Drosophila melanogaster* regular exercise and apolipoprotein B knockdown on abnormal heart rhythm induced by a high-fat diet

**Meng Ding**[1], **Qui Fang Li**[1], **Guo Yin**[1], **Jing Lin Liu**[1], **Xiao Yi Jan**[1], **Ting Huang**[1], **Ai Chun Li**[2]*, **Lan Zheng**[1]*

**1** Key Laboratory of Physical Fitness and Exercise Rehabilitation of Hunan Province, Hunan Normal University, Changsha, China, **2** Faculty of Physical Education, Hainan Normal University, Haikou, China

\* Lanzheng@hunnu.edu.cn (LZ); 739042641@qq.com (ACL)

**Data Availability Statement:** All relevant data are within the manuscript and its Supporting Information files.

## Abstract

Abnormal heart rhythm is a common cardiac dysfunction in obese patients, and its pathogenesis is related to systemic lipid accumulation. The cardiomyocyte-derived *apoLpp* (homologous gene in *Drosophila* of the human apolipoprotein B) plays an important role in whole-body lipid metabolism of *Drosophila* under a high-fat diet (HFD). Knockdown of *apoLpp* derived from cardiomyocytes can reduce HFD-induced weight gain and abdominal lipid accumulation. In addition, exercise can reduce the total amount of *apoLpp* in circulation. However, the relationship between regular exercise, cardiomyocyte-derived *apoLpp* and abnormal heart rhythm is unclear. We found that an HFD increased the level of triglyceride (TG) in the whole-body, lipid accumulation and obesity in *Drosophila*. Moreover, the expression of *apoLpp* in the heart increased sharply, the heart rate and arrhythmia index increased and fibrillation occurred. Conversely, regular exercise or cardiomyocyte-derived *apoLpp* knockdown reduced the TG level in the whole-body of *Drosophila*. This significantly reduced the arrhythmia induced by obesity, including the reduction of heart rate, arrhythmia index, and fibrillation. Under HFD conditions, flies with *apoLpp* knockdown in the heart could resist the abnormal cardiac rhythm caused by obesity after receiving regular exercise. HFD-induced obesity and abnormal cardiac rhythm may be related to the acute increase of cardiomyocyte-derived *apoLpp*. Regular exercise and inhibition of cardiomyocyte-derived *apoLpp* can reduce the HFD-induced abnormal cardiac rhythm.

## Introduction

Obesity increases the risk of cardiovascular disease, increases blood lipids and impairs exercise capacity [1]. *Drosophila* is a good model for metabolism and diet-related diseases and its core pathway for regulating energy has been highly conserved in evolution [2]. It is similar to mammals in that excessive chemical energy in the form of lipids and glycogen are stored in the fat

**Funding:** National Natural Science Foundation of China (project number: 32071175); the Hunan Province Graduate Education Innovation Project and Professional Ability Enhancement Project Fund (Project Number: CX20200533); China Postdoctoral Science Foundation funded project (Project Number: 2017M622580). The funder had no role in study design, data collection and analysis, decision to publish, or preparation of the manuscript.

**Competing interests:** The authors have declared that no competing interests exist.

body [3]. Fat bodies are similar to human adipose tissue and have a liver function [3]. They store lipids in lipid droplets (an organelle). *Drosophila* females have more triglyceride storage than males and triglycerides respond more slowly to lipolysis stimuli. In addition, the female *Drosophila* body is larger and easier to observe and dissect. When the flies are maintained on a high-fat diet (HFD) the obesity phenotype appears. This reduces heart contractility, blocks conduction, increases structural disease, and causes an abnormal heart rhythm [4]. Therefore, an important way to manage HFD-induced abnormal cardiac rhythm is by controlling lipid metabolism.

Atrial fibrillation (AF) in humans is a clinically severe arrhythmia, similar to the fibrillation of the *Drosophila* myocardium. A HFD in *Drosophila* can increase the heart rate and arrhythmia index and cause fibrillation [5]. Fibrillation of the Drosophila heart may result from lipotoxic damage related to the insulin-TOR signal, which is moderate reduction in insulin-TOR signaling prevents HFD-induced obesity and cardiac dysfunction [4]. Apolipoprotein B (apoB) is related to lipid metabolism. It is the carrier of very-low-density lipoprotein (VLDL) and chylomicrons (CM) [6]. In *Drosophila*, HFD feeding leads to overexpression of *apoLpp* (apolipoprotein B homologous gene) in the heart and lipid accumulation while inhibiting the expression of cardiomyocyte-derived *apoLpp* can reduce the effects of an HFD [7]. Some studies have reported the role of *Drosophila apoLpp* in lipid metabolism but the effects of cardiomyocyte-derived *apoLpp* on abnormal heart rhythm under an HFD are unclear.

Exercise is an effective means for treating obesity and improving cardiovascular health [8, 9]. Obesity increases the incidence of atrial fibrillation, but this can be reduced by exercise [10]. In *Drosophila*, moderate exercise can reduce HFD-induced lipid toxicity damage to the heart, reduce the occurrence of fibrillation, and improve heart rhythm and contractile function [5, 11, 12]. However, the effects of regular exercise combined with cardiomyocyte *apoLpp* knockdown on abnormal heart rhythm under HFD are unclear. This study documented the effects of regular exercise and *apoLpp* knockdown in cardiomyocytes on abnormal heart rhythms in *Drosophila melanogaster*.

## Materials and methods

### Fly stocks and diet

We obtained wild-type $W^{1118}$ and *Hand-Gal4* ($W^{[1118]}$; $P\{y^{[+t7.7]}$ $w^{[+mC]}$ = GMR88D05-GAL4} attP2) *Drosophila melanogaster* from the fruit fly breeding room of Hunan Provincial Key Laboratory of Physical Fitness and Sports Rehabilitation of Hunan Normal University; *UAS-apoLpp RNAi* ($y^1$ $sc^*$ $v^1$ $sev^{21}$; $P\{TRiP.HMS00265\}attP2/TM3$, $Sb^1$) were purchased from the Bloomington Drosophila Stock Center, Bloomington, IN, USA. Females of the *UAS-apoLppRNAi* strain were crossed with males of $W^{1118}$ and F1 generation virgin flies were collected as a control group. Females of the *UAS-apoLppRNAi* strain were crossed with males of *Hand-Gal4*, and the F1 generation virgin flies were collected for the intervention group (excluding the female flies with short bristles). All of the flies are placed in an HWS intelligent incubator and maintained at 25°C, 50% relative humidity (RH), and a 12:12 h (L:D) photoperiod. Unless otherwise stated, all flies used in the experiment were females. Their larger size made dissection easier and they were used in previous imaging work in our lab [13]. Normal food (NF) was a combination of yeast, corn starch and molasses. An HFD was made with 30% coconut oil mixed with 70% NF [4].

### HFD feeding regime

Virgin flies were collected and placed on NF medium for five days of aging. They were then transferred to fresh HFD medium for five days [4].

## Regular exercise protocols

Based on the negative geotaxis behavior of flies [14], we developed a *Drosophila* exercise device, which rotates a glass tube to induce the flies to climb upwards [11]. The experiment used HFD fed flies that also exercised. The exercise device rotated 180˚ every 24 s, and the exercise duration was 1.5 h daily for five days. This exercise intensity was sufficient to reduce the body lipid level of the flies [11]. The flies are placed in an incubator at 25˚C with 50% RH and a 12:12 h (L:D) photoperiod. Exercise was conducted regularly in a temperature-controlled room at 25˚C. A sponge plug was adjusted before exercise so that all flies had the same movement distance in the glass tube. In the exercise training intervention, flies spent 1.5 h in a glass tube without food. Flies in the exercise-trained control group were also placed in a glass tube without food for 1.5 h (they were in the same environment as the control group but had no exercise training). We ensured that all flies were in the same environment to avoid influencing their feeding rate [15].

## qRT-PCR

A total of 60 flies were dissected in PBS to open the abdominal cuticle and expose the heart. A vacuum pump was used to suck up the excess impurities around the heart (fat and arterial muscle attached to the heart, as well as the surrounding kidney cells). We put 60 isolated hearts into 1 mL of TRIzol reagent lysis solution to extract RNA, and added 10 μL of RNase Free dH2O to dissolve them. We used a Takara reverse transcription kit for reverse transcription. The cDNA was diluted to 40 μL with RNase Free dH2O and stored at −20˚C. We used a Takara qRT-PCR kit to perform real-time quantification on a Bio-Rad 96-well fluorescent quantitative RT-PCR instrument (ABI7300, Applied Biosystems, USA). We determined the relative gene expression level by comparing the CT method. *apoLpp* primer: F = 5′-AATTC GCGGATGGTCTGTGT-3′; R = 5′-GCCCCTTAGGGATAGCCTTT-3′. Gapdh primer: F = 5′-GCGTCACCTGAAGATCCCAT-3′; R = 5′-GAAGTGGTTCGCCTGGAAGA-3′.

## Semi-intact *Drosophila* preparation and image analysis

We measured cardiac function parameters using previously described methods [16, 17]. Fly-Nap (Sangon Biotech, Shanghai, China) was used to anesthetize the flies and then we performed semi-exposed heart dissection. We used an EM-CCD high-speed camera to capture the fly heartbeat (130 fps, 30-s video), semi-automated optical heartbeat analysis software (SOHA) to quantify the heart rate (HR), arrhythmia index (Arrhythmia Index, AI), and fibrillations (FL).

## Body weight and triglycerides assay

We used an electronic microbalance (Uni bloc, AUW220D, Japan) to weigh the flies and we recorded the weight of each fly for analysis (S1 Table).

We used the Insect TG ELISA Kit (mlbio, China) to measure the TG concentration according to manufacturer instructions. For quantification of TGs in whole flies, flies (15 per genotype) were weighed and homogenized in PBS containing 0.1% Triton-X100 in an amount (μl) that was 8 X the total weight of the flies (μg). Then, centrifugation was used and the supernatant was obtained. Fifty microliters of standard or sample were added to the appropriate wells. Blank wells had nothing added. One hundred microliters of enzyme conjugate were added to standard wells and sample wells except for the blank well; they were covered with an adhesive strip and incubated for 60 minutes at 37˚C. The microtiter plate was washed four times then, Substrate A (50 μl) and Substrate B (50 μl) were both added to each well, mixed gently, and

incubated for 15 minutes at 37˚C (while being protected from light). Following this, 50 μl stop solution was added to each well. The color in the wells should change from blue to yellow. If the well color was green or the color change does not appear uniform, the plate was gently tapped to ensure thorough mixing. The optical density (OD) was read at 450 nm using a microtiter plate reader within 15 minutes. Using the OD value of the measured standard product as the abscissa and the standard product's concentration value as the ordinate, the standard curve was drawn to obtain the linear regression equation. The OD value of the sample was substituted into the equation to calculate the concentration of the sample. The triglyceride standard curve obtained is presented in the (S1 Fig).

## Negative geotaxis assay

Negative geotaxis is an innate escape response of *Drosophila* [18]. To test the climbing ability of flies, we transferred the flies to a 20-cm-long glass tube divided into nine areas and sealed with a sponge plug to leave 18 cm as the fly climbing area. Each area had a fixed score. The tube was shaken every 30 s to make the flies fall to the bottom of the tube and this was repeated three times. Before shooting, we allowed the flies to acclimate in the glass tube for 10 min. We then used a digital camera to take a video of flies climbing, capture the climbing image after 8 s, and calculate the total score. Climbing index = total score/total number (S2 Table).

## Quantification of ORO staining

Photoshop software was used to quantify the intensity of the ORO staining [19]. The all of the eggs and intact intestines of flies were removed in PBS, fixed with 4% paraformaldehyde for 20 min, and washed twice with PBS. They were dyed with fresh ORO dye solution (6 ml of 0.1% ORO in isopropanol and 4 ml of distilled water) for 20 min and rinsed with distilled water [20]. It should be noted that the heart and other internal organs of the fruit flies must be removed gently to avoid entrainment of fat and reduce the staining effect. A Leica stereomicroscope was used to collect bright-field images. We used Adobe Photoshop to adjust brightness and contrast to clarify the images. And then invert these color photos into black and white images in Photoshop.These color images were inverted (image > adj. > invert) in Photoshop to obtain black/white images. Intensity was measured using histogram analysis (mean pixels). Five individual sections (using a constant square area) were measured and averaged within each flies. The photos used for analysis are in the (S1 File).

## Statistical analyses

We used SPSS (version 22.0 for Windows; SPSS Inc., Chicago, IL, USA) for statistical analysis, and comparisons between the NF group and other experimental groups used an independent *t*-test. We used two-way analysis of variance (ANOVA) and a least significant difference (LSD) test for analysis between groups (HFD, HE, HFD+KD, and HFD+E+KD). When the variance was not uniform, we used non-parametric tests such as Kruskal–Wallis one-way ANOVA (K sample) for post-adjustment. All of the data are expressed as mean±standard error (X±SEM), $\alpha = 0.05$. $^*p<0.05$, $^{**}p<0.01$, $^{***}p<0.001^*$.

## Ethical statement

This study was approved by the Ethics Committee of Hunan Normal University.

## Results

### Increased dietary fat causes obesity and abnormal heart rhythms in Drosophila

To study the effect of lipid accumulation on cardiac function in *Drosophila*, we first determined the effect of an HFD on the whole-body lipid metabolism in *Drosophila*. In mammals, elevated TG levels are the main risk factor for obesity and metabolic syndrome [21, 22]. Obesity also increases the risk of cardiovascular disease and weakens physical activity [23]. After five days of feeding on an HFD, the flies gained weight (Fig 1A) and the intensity of ORO

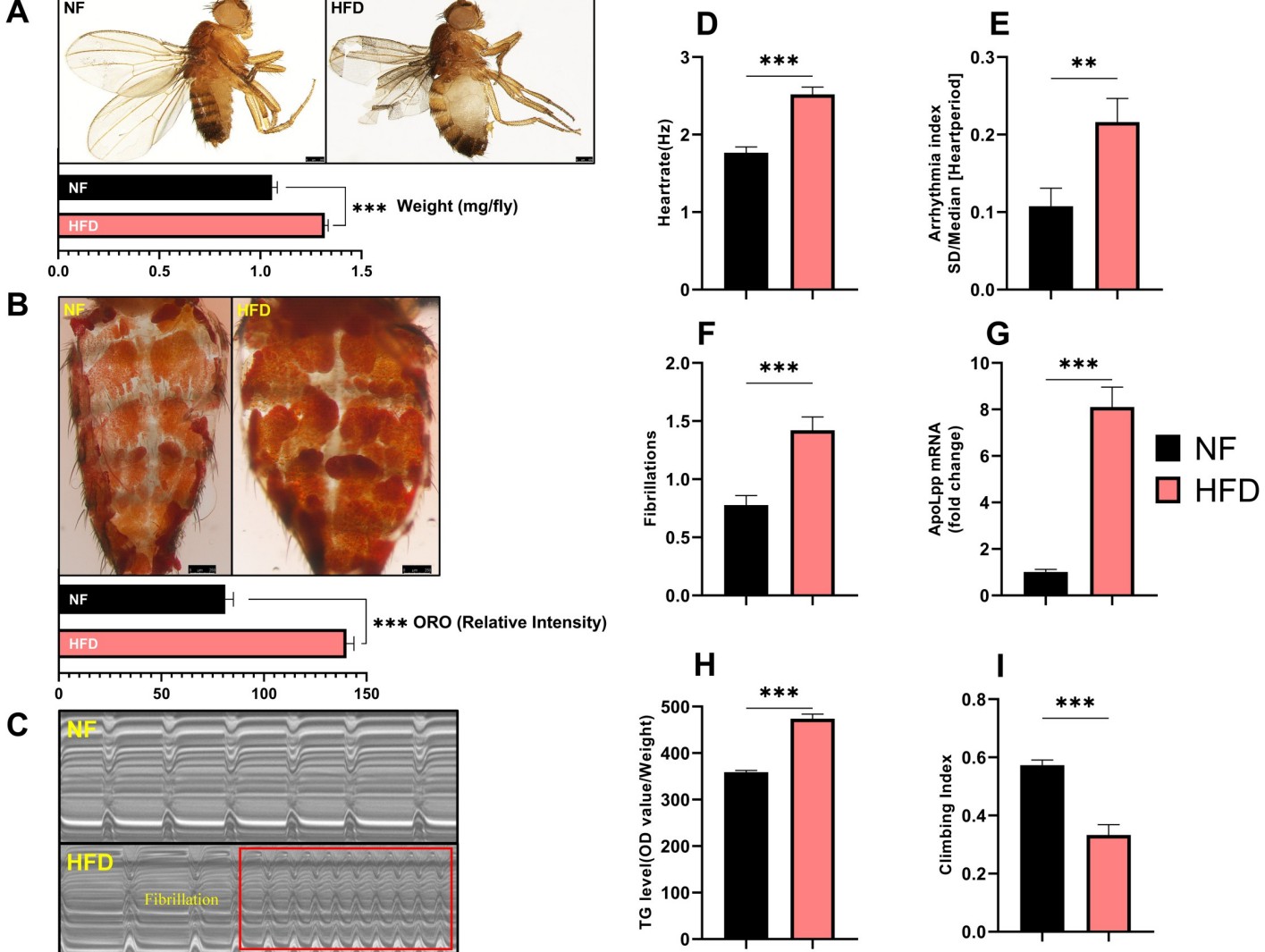

**Fig 1. *UAS-apoLpp RNAi*>W[1118] group was exposed to HFD for 5 days, resulting in obesity and abnormal heart rhythm.** (A) Photographs and body weights of 10-day-old flies. The body weight was obtained by weighing flies on an electronic microbalance, N = 15. The flies in the HFD group were heavier compared to the NF group. (B) ORO staining of the abdomen of flies. Quantification of ORO intensity, N = 5. (B) ORO staining of fly abdomens. Quantification of ORO intensity, N = 5. The intensity of ORO staining in the abdomen of flies in the HFD group was higher, compared to the HF group. (C) *Drosophila* M-mode cardiogram, the red rectangle represents fibrillation, and the interception length is 10 s (This refers to the length of the electrocardiogram of 0–10 s). (D-F) M-Mode analysis. Quantification of the fly heart rate, arrhythmia index, and fibrillation, N = 30. The heart rate, arrhythmia index, and fibrillation of flies in the HFD group were higher than those in the NF group. (G) The relative expression level of *apoLpp* in cardiomyocytes of flies. The *apoLpp* in the cardiomyocytes of flies in the HFD group was significantly greater than in the NF group. The samples included at least 60 isolated hearts. (H) Whole-body TG levels in flies. The whole-body TG level of the HFD group was significantly higher than that of the NF group. N = 5, repeated three times. (I) *Drosophila* climbing index. The climbing index of the HFD group was significantly less than that of the NF group. The climbing index = number of flies at the top/total number of flies, N = 50, repeated three times. The detailed method is given in the (S2 Table).

staining increased (Fig 1B). Furthermore, the whole-body TG level increased (Fig 1H), and the climbing index decreased (Fig 1I). These results demonstrated that HFD feeding can induce obesity in *Drosophila*.

Obesity is an important risk factor for atrial fibrillation. Atrial fibrillation is the most common persistent arrhythmia and can cause morbidity and mortality [24]. To detect whether the hearts of obese *Drosophila* were abnormal, we performed semi-intact preparations and M-mode analysis. Compared with the NF group, the heart rate of flies in the HFD group increased (Fig 1D), and the arrhythmia index and increased fibrillation increased (Fig 1E and 1F). These results indicate that HFD flies have abnormal heart rhythms and fibrillation (Fig 1C). In addition, apolipoprotein B is closely correlated to the occurrence of cardiovascular disease. When the level of apoB increases, individuals have an increased risk of cardiovascular disease, while reducing the level of apoB can reduce cardiovascular disease [25, 26]. In *Drosophila*, apoB plays an important role in controlling systemic lipid metabolism [7]. To determine whether HFD increased the expression of *apoLpp*, we detected *apoLpp* levels in the hearts of HFD *Drosophila*. The expression level of *apoLpp* in the heart of HFD *Drosophila* significantly increased, compared with the level in the NF group (Fig 1G). These results indicate that an increase in dietary fat leads to obesity and abnormal heart rhythm in *Drosophila*. The mechanism may be related to the rise in cardiomyocyte *apoLpp*.

## Cardiomyocyte-specific *apoLpp* knockdown can resist HFD-induced obesity and abnormal heart rhythm

It is unclear how obesity induces abnormal heart rhythms. The *apoLpp* is related to systemic lipid metabolism and heart function in *Drosophila* [7]. Therefore, we tested whether cardiomyocyte-derived *apoLpp* is involved in the abnormal heart rhythm caused by obesity. To determine whether knocking down *apoLpp* can resist abnormal heart rhythm caused by obesity, we used Hand-Gal4 to specifically inhibit the expression of cardiomyocyte *apoLpp* (Fig 2G). We first tested whether the increase in whole-body TG levels due to HFD was suppressed in the HFD+KD group of flies. The knockdown of cardiomyocyte *apoLpp* significantly reduced the whole-body TG level of flies (Fig 2H). Compared with the HFD group, the body weight of flies in the HFD+KD group was reduced, and the intensity of ORO staining was reduced (Fig 2A and 2B). These results indicate that reducing the expression of *apoLpp* in cardiomyocytes helps to weaken the obesity caused by HFD. Since it has been demonstrated that HFD can induce an abnormal heart rhythm in *Drosophila*, and the inhibition of *apoLpp* can reduce the obesity caused by HFD, we speculated that the inhibition of *apoLpp* in the cardiomyocytes of flies can reduce the abnormal heart rhythm induced by HFD. We tested whether the inhibition of *apoLpp* in cardiomyocytes can reduce abnormal heart rhythms under HFD conditions. As expected, compared with the HFD group, the heart rate of flies in the HFD+KD group decreased (Fig 2D), and the arrhythmia index (Fig 2E) and fibrillation (Fig 2F) were reduced. The abnormal heart rhythm of flies was improved (Fig 2C). These findings indicate that the inhibition of cardiomyocyte *apoLpp* in *Drosophila* can reduce HFD-induced obesity, which may help reduce the adverse effects of obesity on the heart.

## Regular exercise reduced the expression level of apoLpp mRNA in cardiomyocytes and reversed the abnormal heart rhythm caused by the HFD diet

Appropriate exercise can aid in weight reduction and can reduce the risk of cardiovascular events [9, 27]. Exercise training can increase cardiac lipid metabolism and prevent

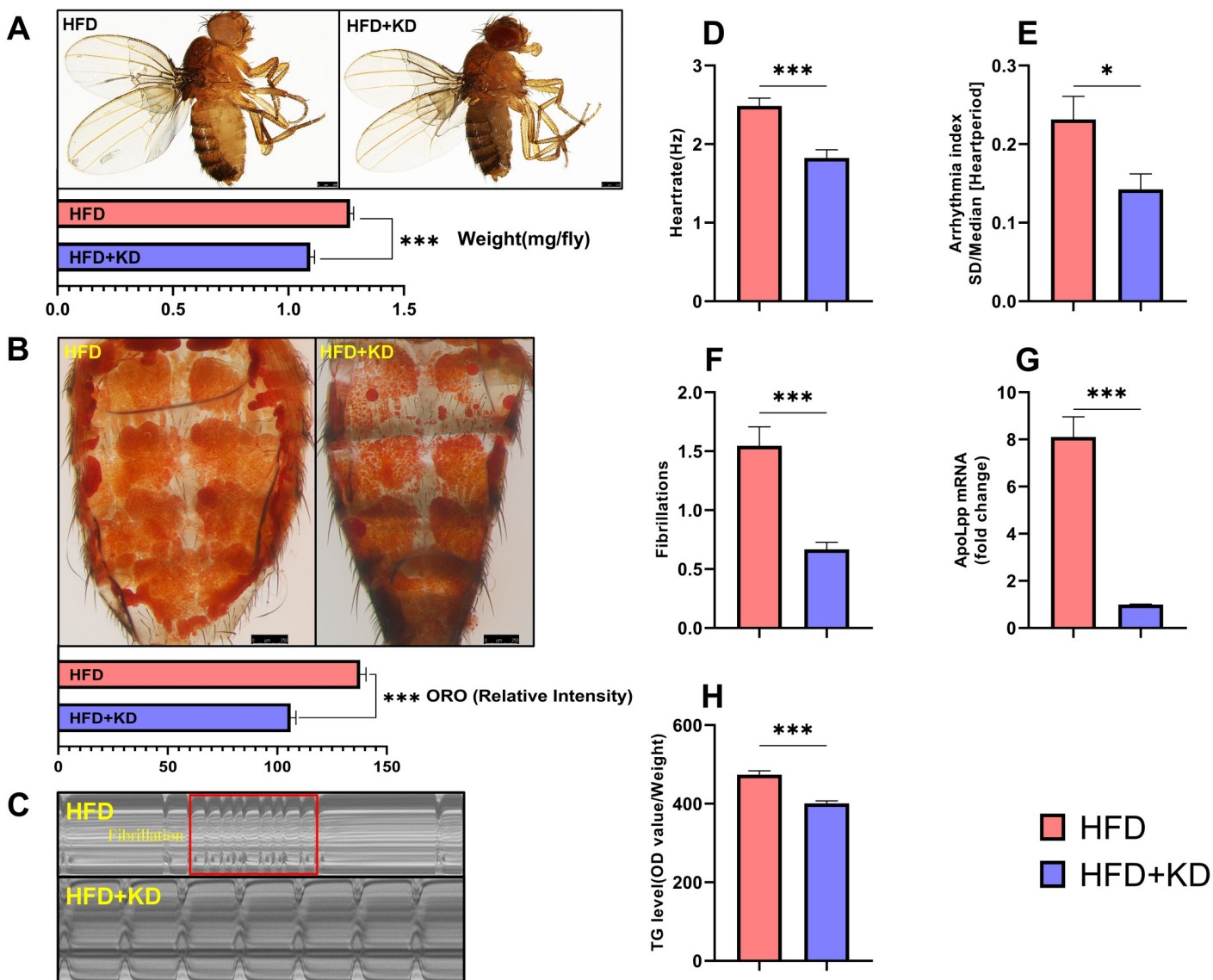

**Fig 2. Under HFD feeding, inhibition of *apoLpp* in cardiomyocytes can reduce obesity and abnormal heart rhythm.** (A) Photographs and body weights of 10-day-old flies. The body weight was obtained with an electronic microbalance, N = 15. The flies in the HFD+KD group were lighter compared to the HFD group. (B) ORO staining of the fly abdomens. Quantification of ORO intensity, N = 5. The intensity of ORO staining in the fly abdomens in the HFD+KD group was reduced, compared with the HFD group. (C) *Drosophila* M-mode cardiogram, the red rectangle represents fibrillation, and the interception length was 10 s (This refers to the length of the electrocardiogram of 0–10 s). (D-F) M-Mode analysis. Quantification of fly heart rate, arrhythmia index, and fibrillation, N = 30. The heart rate, arrhythmia index, and fibrillation of flies in the HFD+KD group were lower than in the HFD group. (G) The relative expression level of *apoLpp* in the cardiomyocytes of flies. The *apoLpp* in cardiomyocytes of flies in the HFD+KD group was significantly lower than that in the HFD group. The test samples included at least 60 isolated hearts. (H) Whole-body TG levels in flies. The whole-body TG level of the HFD+KD group was significantly lower than that of the HFD group. N = 5, repeated three times.

cardiovascular disease [28]. However, it is unclear if regular exercise can improve abnormal heart rhythm by changing *apoLpp* levels in cardiomyocytes. To test this, we made flies exposed to an HFD exercise regularly. The expression of *apoLpp* in cardiomyocytes of flies in the HE group was significantly lower than that in the HFD group (Fig 3H). In addition, after regular exercise, flies lost body weight, the intensity of ORO staining decreased and the level of TG in the whole-body decreased (Fig 3A, 3B and 3G). To further determine whether regular exercise

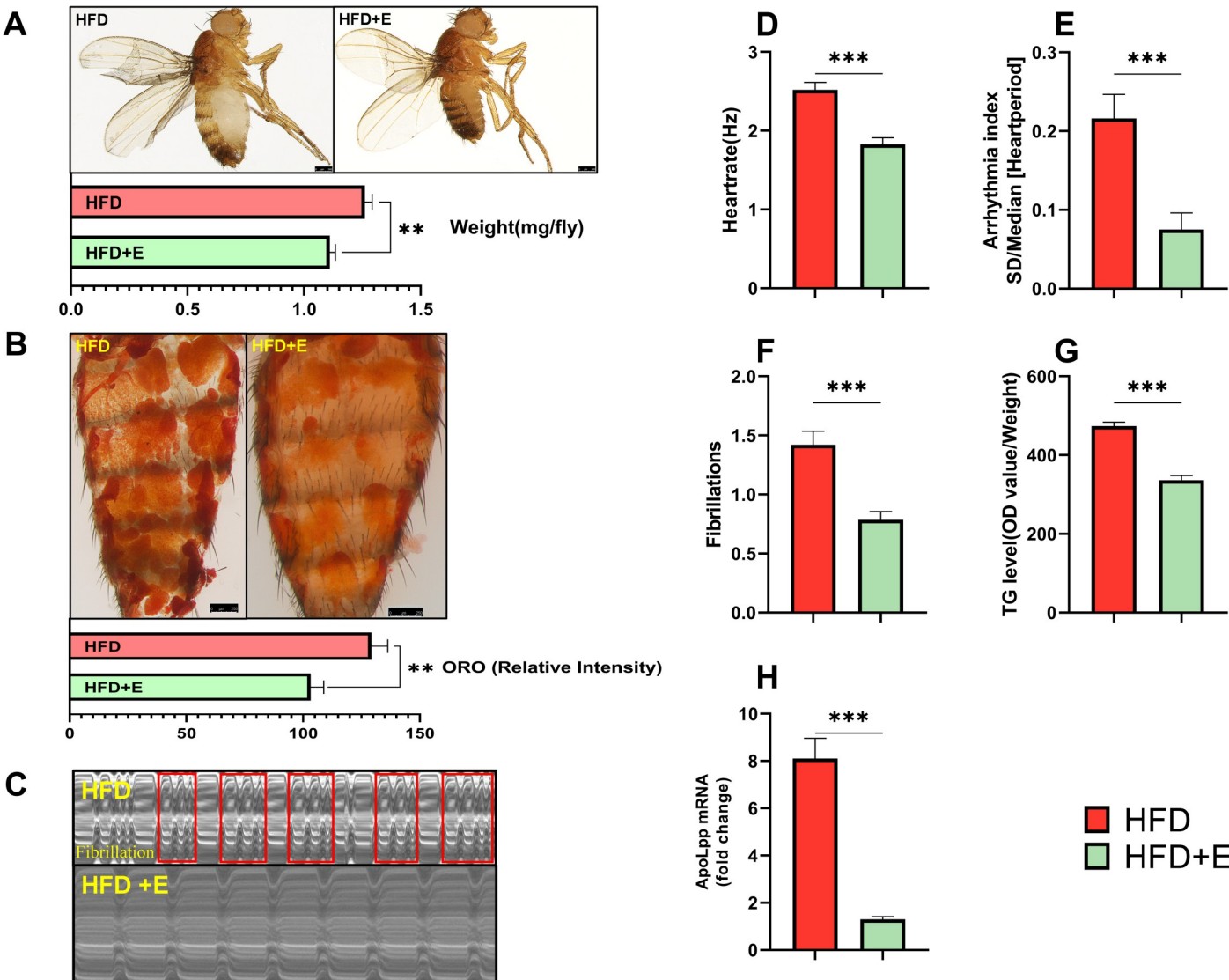

**Fig 3. Regular exercise can reduce the expression level of *apoLpp* mRNA in cardiomyocytes and reverse the abnormal heart rhythm caused by the HFD diet.**
(A) Photographs and body weights of 10-day-old flies. The body weight was obtained using an electronic microbalance, N = 15. The body weight of flies in the HE group was lower than the weight of the HFD group. (B) ORO staining of the abdomen of flies. Quantification of ORO intensity, N = 5. The intensity of ORO staining in the abdomen of flies in the HFD+E group was reduced compared to the HFD group. (C) *Drosophila* M-mode cardiogram, the red rectangle represents fibrillation, and the interception length is 10 s (This refers to the length of the electrocardiogram of 0–10 s). (D-F) M-Mode analysis. Quantification of the fly heart rate, arrhythmia index and fibrillation, N = 30 and 20. The heart rate, arrhythmia index, and fibrillation of flies in the HFD+E group were lower than those in the HFD group. (G) Whole-body TG levels in flies. The whole-body TG level of the HFD+E group was significantly lower than that of the HFD group. N = 5, repeated three times. (H) Relative expression level of *apoLpp* in cardiomyocytes of flies. The *apoLpp* in cardiomyocytes of flies in the HFD+E group was significantly lower than that in the HFD group. The samples included at least 60 isolated hearts.

can reduce the abnormal heart rhythm caused by obesity, we used M-mode to analyze the heart function of flies exposed to HFD. Compared with the HFD group, the heart rate of flies in the HE group decreased (Fig 3D) and the arrhythmia index (Fig 3E) and fibrillation (Fig 3F) also decreased. The abnormal heart rhythm of the flies was improved (Fig 2C). These findings show that regular exercise can downregulate the expression level of *apoLpp* in cardiomyocytes, thereby reducing the abnormal heart rhythm caused by obesity.

## Regular exercise combined with the knockdown effect of cardiomyocytes apoLpp has a significant effect on obesity-induced abnormal heart rhythm

The question the above findings raise is whether regular exercise and the knockdown effect of cardiomyocytes *apoLpp* have a significant effect on abnormal heart rhythm caused by obesity. To make it easier to observe the difference, we compare the data of the HFD control group with the other three groups side by side. We found that the body weight of the HFD group was higher than the other three groups. In addition, compared with the NF and HFD+KD groups, the HFD+E+KD group has no significant difference in the body weight of the flies. (Fig 4A). For ORO staining, the ORO staining intensity of HFD+KD and HFD+E+KD groups was significantly lower than that of HFD group, but they were higher than NF group. (Fig 4B). These data indicate that the knockdown of *apoLpp* in cardiomyocytes or the intervention of knockdown, combined with exercise, can resist the increase in body weight of flies due to HFD. In addition, although the above-mentioned intervention method can control body weight, it is less effective in controlling abdominal fat, because the intensity of ORO staining was higher than that of the NF group. M-mode data showed that the heart rate, arrhythmia index and fibrillation of the HFD group were significantly higher than those of the other three groups. In addition, compared with the NF group, there were no significant differences in heart rate, arrhythmia index, and fibrillation in the HFD+E+KD group (Fig 4C–4F). Compared to the NFD+KD group, the HFD+E+KD group was not different in heart rate and fibrillation (Fig 4C, 4D and 4F), while the arrhythmia index was significantly lower (Fig 4E). These results indicate that regular exercise, combined with the knockdown of cardiomyocytes *apoLpp*, significantly impacts the arrhythmia index. In addition, the expression level of *apoLpp* mRNA in flies cardiomyocytes in the HFD group was significantly higher than that of the other three groups. And compared with flies in the NF and HFD+KD groups, the expression level of *apoLpp* in the heart of the HFD+E+KD group was significantly reduced (Fig 4G), and the level of TG in the whole-body was reduced (Fig 4H). These results indicate that regular exercise, in combination with *apoLpp* knockdown in cardiomyocytes, can further reduce the expression of *apoLpp* in cardiomyocytes. This reduces the increase in TG levels of flies under HFD conditions, even lower than normal. In summary, these data indicate that in addition to regular exercise, cardiomyocyte *apoLpp* is required to improve HFD-inducted abnormal heart rhythm requires, because *apoLpp* is more flexible during HFD feeding.

## Discussion

### Effects of high-fat diet on the expression of apoLpp in the cardiomyocytes of Drosophila

The imbalance between energy intake and energy expenditure leads to obesity and obesity can cause abnormalities in cardiovascular hemodynamics, heart shape and ventricular function [29]. In mice, an HFD period can cause abnormal lipid metabolism, which is mainly manifested as abnormal blood lipid/glycemia, insulin resistance, and damage to heart function [30]. In *Drosophila*, the damage to heart function caused by obesity induced by an HFD is evolutionarily conserved [31]. Other studies have found that an HFD can cause increased expression of *apoLpp* in *Drosophila* myocardial cells, weakened myocardial contractility, abnormal heart rhythm and remodeling of the heart structure [4]. After exposure to HFD for five days, the fly whole-body TG level increased, body weight increased, ORO staining intensity increased and cardiac function became abnormal (rapid heart rate, arrhythmia, fibrillation). We also found that an HFD-induced abnormal heart rhythm and fibrillation are related to the increased expression of *apoLpp* in cardiomyocytes.

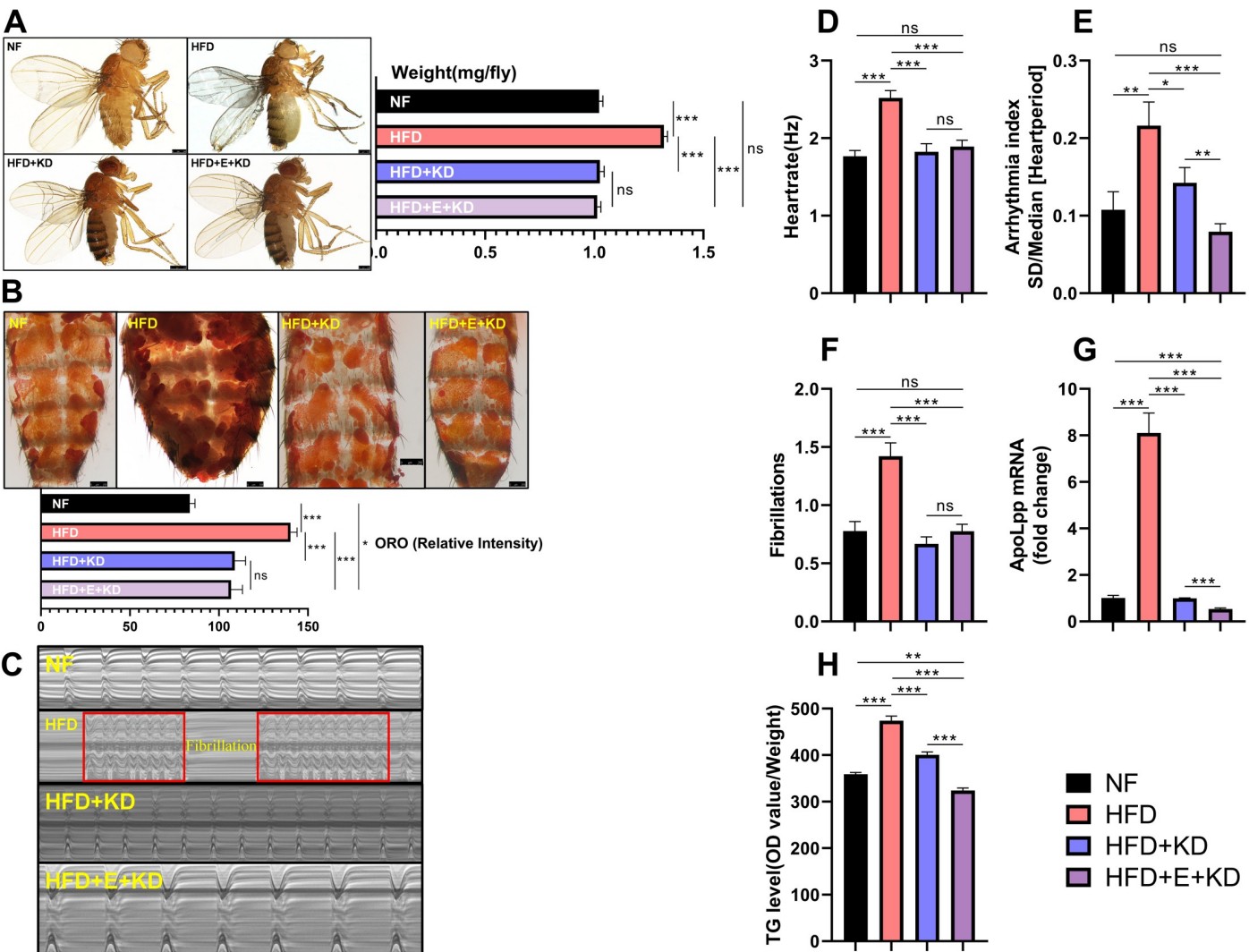

**Fig 4. Combined effect of regular exercise and *apoLpp* knockdown in cardiomyocytes.** (A) Photographs and body weights of 10-day-old flies. The body weight was obtained using an electronic microbalance, N = 15. The body weight of flies in the HFD group was significantly higher than the other three groups. (B) ORO staining of the abdomen of flies. Quantification of ORO intensity, N = 5. The intensity of ORO staining of flies in the HFD group was significantly higher than that of the other three groups. (C) *Drosophila* M-mode cardiogram and the interception length was 10 s (This refers to the length of the electrocardiogram of 0–10 s). (D-F) M-Mode analysis. Quantification of fly heart rate, arrhythmia index and fibrillation, N = 30. The heart rate, arrhythmia index and fibrillation of the three groups of NF, HFD+KD and HFD+E+KD were significantly lower than those of the HFD group. Compared with the HFD+KD group, the HFD+E+KD group had no significant difference in heart rate and fibrillation, but the arrhythmia index was significantly lower. (G) The relative expression level of *apoLpp* in cardiomyocytes of flies. The expression level of *apoLpp* mRNA in cardiomyocytes of HFD group was significantly higher than that of the other three groups. In addition, compared with HFD+KD, the expression of *apoLpp* mRNA in cardiomyocytes of HFD+E+KD group was significantly reduced. The samples included at least 60 isolated hearts. (H) Whole-body TG levels in flies. The whole-body TG level in the HFD group was significantly higher than the other three groups. In addition, compared with the NF and HFD+KD groups, the HFD+E+KD group's whole-body TG levels were significantly lower. N = 5, repeated three times.

Elevated levels of apolipoprotein B are a sign of metabolic syndrome, including obesity, diabetes, and heart disease [32]. The function of *apoLpp* in *Drosophila* is similar to that of apolipoprotein B in mammals (its receptor is homologous to mammals), and they can transport lipids and sterols to surrounding tissues to control the storage and mobilization of neutral lipids in cells [33, 34]. *Drosophila* cardiomyocyte *apoLpp* plays an essential role in controlling systemic lipid metabolism and actively responds to HFDs [7]. Therefore, the effect of an HFD on the

*apoLpp* of cardiomyocytes is highly correlated, but the relationship between *apoLpp* and abnormal heart rhythm induced by an HFD is still unclear.

## The effects of cardiomyocytes apoLpp knockdown in HFD flies on abnormal heart rhythm

An HFD can cause cardiac lipid toxicity and induce cardiac dysfunction, mainly including arrhythmia, fibrillation, and weakened contractility [5, 11, 12, 31]. However, it is unclear whether abnormal heart rhythm induced by an HFD is affected by *apoLpp*. To further explore the abnormal heart rhythm induced by an HFD, we knocked down the *apoLpp* gene in the cardiomyocytes of flies. The *apoLpp* expression level of fly cardiomyocytes in the HFD+KD group was reduced by 87.7%, compared with the HFD group, which indicated the successful construction of the knockdown strain. The M-type results showed that the heart rate was slowed, the arrhythmia index was reduced, and fibrillation did not occur, reflecting the improvement of heart function. These data indicate that the knockdown of *apoLpp* in cardiomyocytes can resist the abnormal heart rhythm caused by HFD. Moreover, in the flies exposed to HFD, after *apoLpp* knockdown in cardiomyocytes, whole-body triglyceride levels were significantly reduced, body weight decreased, and ORO staining intensity decreased. These data indicate that the knockdown of cardiac *apoLpp* can significantly improve lipid accumulation caused by a HFD, which is consistent with the results of other studies [7]. Knockdown of *apoLpp* in cardiomyocytes can improve abnormal heart rhythm, which may be caused by changes in the lipid overload environment [35].

## Effects of apoLpp knockdown in fly cardiomyocytes, combined with regular exercise, on abnormal heart rhythm

Physical exercise can enhance heart function and reduce diabetic cardiomyopathy, coronary heart disease, and heart failure [36, 37]. Both rats and mice can improve heart disease induced by high glucose or high fat through regular exercise [38, 39]. Drosophila has shown similar results to mammals in terms of heart damage caused by high fat or high sugar [40, 41]. In addition, exercise also has many benefits for flies. Examples include lipotoxic cardiomyopathy and improvement of heart rhythm [11, 12], as well as resistance to age-related degenerative changes in the heart. Previous studies showed that exercise in aged flies can effectively reduce the occurrence of fibrillation [11]. Regular exercise of flies provides partial resistance to the heart function damage induced by an HFD but it is still unclear how regular exercise combined with knockdown of *apoLpp* gene in cardiomyocytes affects arrhythmia. To clarify this issue, we performed regular exercises on the flies exposed to the HFD. Compared to the respective control group, the expression of *apoLpp* in the cardiomyocytes of flies in the exercise group was significantly reduced. In addition, whole-body triglyceride levels, body weight, and intensity of ORO staining were reduced. These data demonstrate that regular exercise can reduce the expression of *apoLpp* in cardiomyocytes caused by an HFD, thereby resisting obesity. M-mode results also showed that regular exercise can reduce heart rate, reduce arrhythmias and reduce fibrillation. In addition, the *Drosophila* cardiogram indicated that the HFD caused more serious arrhythmia in flies such as fibrillation. When flies exposed to HFD performed regular exercise, there was no fibrillation, and arrhythmia improved. When flies exposed to HFD had *apoLpp* knocked down in the cardiomyocytes, fibrillation also did not occur. In addition, regular exercise combined with the knockdown of cardiomyocytes *apoLpp* can further improve the arrhythmia of flies under HFD. This was manifested as a decrease in heart rate, an increase in the cardiac cycle and a decrease in arrhythmia index. Therefore, not only the knockdown effect of *apoLpp*, but also regular exercise can drive the expression of

*apoLpp*, which in turn affects the arrhythmia induced by an HFD. This study also found that, compared with flies exposed to an HFD and knocked down *apoLpp* in cardiomyocytes, regular exercise resulted in lower levels of *apoLpp* in cardiomyocytes, and a slight decrease in heart rate, arrhythmia index, and fibrillation. These results indicate that the benefit of regular exercise on abnormal arrhythmia requires cardiomyocyte *apoLpp*, because cardiomyocyte *apoLpp* RNAi is more flexible during HFD. It is worth noting that the feeding rate is an important reference for studying metabolic regulation [42]. Although HFD itself has a profound impact on feeding rate, it does not affect the establishment of the HFD model in this study. In addition, exercise can also affect food intake, such as overeating after exercise [43]. In the future, it will be interesting to explore the effect of Drosophila exercise-regulated feeding rate on lipid metabolism.

Other studies have reported that ovarian triglyceride levels represent only a small part of whole-body triglyceride levels [44, 45] in females. The accumulation of lipids is also not negligible in reproduction, because the ovary is an important area of fat accumulation in female *Drosophila*. The relationship between lipid metabolism in the ovary of female *Drosophila* and heart function remains unclear. Considering the shuttle mechanism of lipoproteins, when cardiomyocytes *apoLpp* is specifically knocked down, it will not lead to a sudden decrease of apolipoprotein B. This may be because apolipoprotein B from the fat body enters the heart through hemolymph circulation. Although this subtle difference is interesting, it does not affect the contribution of inhibiting *apoLpp* in cardiomyocytes to the total amount of *apoLpp* in the whole-body. In summary, this study shows that *apoLpp* reduction may be involved in exercise-induced protection against HFD.

## Supporting information

**S1 Fig. Standard curve diagram of *D. melanogaster* whole-body triglyceride detection.**
(TIF)

**S1 Table. Data from all weight panels in this study.**
(XLSX)

**S2 Table. *Drosophila* climbing index.**
(XLSX)

**S1 File. ORO stained photo of flies abdomen.**
(ZIP)

## Acknowledgments

We thank Karen Ocorr and Rolf Bodmer (Sanford Burnham Institute of Neuroscience and Aging Research Center) for supporting semi-automatic optical echocardiographic analysis software. We also thank the School of Physical Education, Hunan Normal University, Bloomington *Drosophila* Stock Center, and the Vienna *Drosophila* RNAi Center for fly stocks. We thank LetPub (www.letpub.com) for its linguistic assistance during the preparation of this manuscript.

## Author Contributions

**Conceptualization:** Meng Ding, Ai Chun Li, Lan Zheng.

**Data curation:** Meng Ding, Ting Huang.

**Formal analysis:** Meng Ding.

**Funding acquisition:** Lan Zheng.

**Investigation:** Meng Ding, Ting Huang.

**Methodology:** Meng Ding, Jing Lin Liu, Ai Chun Li.

**Project administration:** Meng Ding, Jing Lin Liu, Xiao Yi Jan.

**Resources:** Meng Ding, Qui Fang Li.

**Software:** Meng Ding, Qui Fang Li, Jing Lin Liu, Xiao Yi Jan, Ting Huang.

**Supervision:** Meng Ding, Ting Huang.

**Validation:** Meng Ding, Guo Yin, Ting Huang.

**Visualization:** Meng Ding, Qui Fang Li, Guo Yin, Xiao Yi Jan.

**Writing – original draft:** Meng Ding.

**Writing – review & editing:** Meng Ding.

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
