## [Decision Letter · Decision Letter 0]

5 Aug 2021

PONE-D-21-21284

The effect of regular exercise and apolipoprotein B knockdown on abnormal cardiac rhythm induced by HFD

PLOS ONE

Dear Dr. Zheng,

Thank you for submitting your manuscript to PLOS ONE. After careful consideration, we feel that it has merit but does not fully meet PLOS ONE’s publication criteria as it currently stands. Therefore, we invite you to submit a revised version of the manuscript that addresses the points raised during the review process.

We look forward to receiving your revised manuscript.

Kind regards,

Girish C. Melkani, Ph.D

Academic Editor

PLOS ONE

Journal Requirements:

2. When reporting the results of qualitative research, we suggest consulting the COREQ guidelines  or other relevant checklists listed by the Equator Network, such as the SRQR, to ensure complete reporting (http://journals.plos.org/plosone/s/submission-guidelines#loc-qualitative-research). Moreover, please provide the interview guide used as a Supplementary File.

This study was funded by the Instituto de Salud Carlos III (www.isciii.es) PI15CIII/00047 to TBH. The funder had no role in study design, data collection and analysis, decision to publish, or preparation of the manuscript.

Additional Editor Comments (if provided):

I have received assessments your manuscript from two external reviewers and I completely agree with their assessment. Therefore, your manuscript will require a major revision. The subject area is quote interesting however, the reviewers has raised reasonable questions/concerns. All these questions/concern should be addressed in the revised version for the consideration in PLoS ONE. In addition, the authors are re-using the same images for one than one panel (Fig. 1 A and B, HC and Fig. 2 A and B HC). They should use another representative image in each figure panel.

Reviewers' comments:

Reviewer's Responses to Questions

**Comments to the Author**

1. Is the manuscript technically sound, and do the data support the conclusions?

Reviewer #1: Partly

Reviewer #2: Partly

2. Has the statistical analysis been performed appropriately and rigorously? 

Reviewer #1: No

Reviewer #2: Yes

3. Have the authors made all data underlying the findings in their manuscript fully available?

Reviewer #1: Yes

Reviewer #2: Yes

4. Is the manuscript presented in an intelligible fashion and written in standard English?

Reviewer #1: No

Reviewer #2: Yes

5. Review Comments to the Author

Reviewer #1: Manuscript compares the effect of heart-specific apoLpp knockdown and the effect of exercise on consequences of a high fat diet. Although the effects of HFD, apoLpp and exercise on heart performance have all been examined previously, it is of some interest to look at the combination. There are several significant problems with the study that reduce enthusiasm, including genetic background issues, lack of controls for female-specific dietary responses to egg laying, and inappropriate use of Oil Red O as a lipid quantitation assay.

Major:

1. Not enough attention has been paid to rigor in the control of genetic background. Comparisons are made between outcrossed animals and isogenic control animals without recognizing possible effects of hybrid vigor on phenotypes.

2. Rescue experiments are compared to each other (e.g. exercise to apoLpp RNAi) without including comparisons to wild type. This makes it impossible to quantitate the extent of rescue.

3. No attention has been given to possible effects of any of the treatments on feeding rate.

4. Although experiments are done in females, no attention has been given to effects of reproduction on lipid accumulation. This is an essential point to address, because eggs are a substantial subset of the lipid accumulation in any female.

5. No rationale is given for limiting experiments to females.

6. Figures showing representative flies are clearly of different lengths and overall sizes, which surely must complicate assessment of whether they have "flatter abdomens". Abdominal bulging may result from reduced egg deposition and must be controlled for.

7. Oil Red O staining is not a quantitative assay for lipid content.

8. It is a major stretch to say that apoLpp RNAi phenotypes prove that effects of HFD are because of abnormal lipoprotein concentration.

9. Discussion section does not effectively place the work in context, and basically restates the Introduction. Also, it is labelled "Discuss".

Minor:

1. ApoLpp should be spelled out and defined at first mention

2. Manuscript should be heavily edited for grammar.

3. methods descriptions should be in past tense, and some of the statistical methods do not appear quite right. For example, pairwise comparisons after a 2-way ANOVA should use post-hoc adjustment, such as a Tukey test. All the comparisons in the paper appear to be pairwise, so it is not clear why the stats methods describe something different?

4. What is interception length?

5. does "half-exposure surgery" mean semi-intact preparation?

6. Description of Figure 1 refers incorrectly to figure 2 in several places, which is very confusing

7. meaning of "superimposing effect" is not clear

Reviewer #2: Ding et al present an interesting group of studies showing the role of exercise and the apoLpp protein in high-fat-induced cardiac dysfunction. This is a nice short paper with high quality data and will be of interest to those studying the roles of diet and lipid transport on cardiac function. However, a few changes need to be made before I can recommend it for publication in PLoS One.

The triglyceride (TG) assay method is not described correctly- there are no reagents listed and an antibody/substrate approach makes no sense in this protocol because triglycerides are not typical protein antigens. A standard curve should also be used in this assay so the ug TG/mg body weight can be reported. Without the proper methods, the data on TG cannot be evaluated.

It would also be helpful if you described briefly how you isolated hearts; on line 210, it says that qPCR was used to quantify mRNA in cardiomyocytes, but the heart has nephrocytes and aliary muscles that are stuck to it and are sometimes included in the prep.

The heart rate of ~ 2 sec is surprising to me because most research papers have a rate of 2-3 beats per second. Perhaps the units should be Hz instead of seconds? Please double check this.

For Fig 2B’s legend, “there is no accumulation of fat in the abdomen of flies in the HC-KD group” is not supported by the image. Reduced accumulation even looks like a stretch, but seems to be corroborated by the quantitative data in 2H provided that a legitimate TG assay was done.

Figure 3’s title text “Regular exercise improves abnormal heart rhythm by reducing the expression of apoLpp mRNA in cardiomyocytes” is not supported by the data. The authors have only shown that apoLpp is reduced by exercise, not that this is the mechanism by which heart function is improved. For that, I think you’d have to overexpress apoLpp in the heart during exercise and show that heart function differs from the exercised control genotype. This experiment is one I would request if this were a different journal, but isn’t necessary to publish in PLoS One.

Fig 3G is TG and Fig 3H is apoLpp mRNA, but the legend has it the other way around.

With respect to the figure legends for panel A in Figs 1-4, I think it would be more clear if the text read simply “the abdomen is larger” (or smaller) rather than more prominent or flatter. It’s harder to think about how prominence and flatness are being measured and these might mean different things to different people. Large/small is straightforward- it would be even better if you graphed their wet or dry weights, data you might already have from the triglyceride assays. For 4A, remove significant; significant is reserved for statistical significance in my mind. You might say “no dramatic difference” instead. Again in 4A, I would focus on the size (smaller or larger), rather than “morphology,” which is not as easily ascertained from the images provided. There is only one view shown (perhaps the dorsal abdomen is spotted!) and the size is the most apparent difference in all of these panels.

It would be of interest to directly compare control and apoLpp RNAi responses to exercise. This is difficult considering the way the figures are currently set up; perhaps it could be helpful to add a table or expand the Results or Discussion to mention these. I found the term “superimposed” difficult to understand. It could be rephrased as “Cardiac apoLpp is required for (some of) the benefits of exercise” – then you might consider why there is no longer a benefit from exercise. It may be because apoLpp RNAi hearts are more resilient during HF feeding. So cardiac apoLpp knockdown could be metabolically equivalent to exercising… this would be great, like an exercise mimetic. When I compare 3D and 4D, it looks like the heart rate is already an “exercised” value in the RNAi flies without exercise; the same seems to be true for fibrillation rates in 3F vs 4 F.

There are several small mistakes in writing that should be addressed- I will only mention a few here. Genes and mRNAs should be italicized. Line 110 should say reagent and line 111 and elsewhere should say H2O with a subscript #2. The use of female flies appears four times in the Fig 1 legend and three times in Figure 4’s legend and should be compressed to be as concise as possible. You can mention all flies are female early in the methods and results- then maybe in the Discussion (not Discuss, as on line 336) when comparing your findings to other high fat diet studies.

Finally, I am sleep deprived but I found it hard to keep it in my mind that HFD=HC. Why not use HFD throughout all of the text and figures? It would be easier for some people.

6. PLOS authors have the option to publish the peer review history of their article (what does this mean?). If published, this will include your full peer review and any attached files.

Reviewer #1: No

Reviewer #2: No

---

## [Author Response · Author response to Decision Letter 0]

21 Sep 2021

List of Responses

Dear Editors and Reviewers:

Thank you for your letter and for the reviewers’ comments concerning our manuscript entitled “The effect of regular exercise and apolipoprotein B knockdown on abnormal cardiac rhythm induced by HFD” (ID: PONE-D-21-21284). Those comments are all valuable and very helpful for revising and improving our paper, as well as the important guiding significance to our researches. We have studied comments carefully and have made correction which we hope meet with approval. Revised portion are marked in red in the paper. The main corrections in the paper and the responds to the reviewer’s comments are as flowing:

Responds to the editor's comments:

1.We carefully modify the format of the manuscript to ensure that the manuscript meets the style requirements of PLOS ONE.

2.After reviewers’ suggestions, some experiments were added, and qualitative analysis was transformed into quantitative analysis, making the article more objective.

3.This research was funded by the following: National Natural Science Foundation of China (project number: 32071175); the Hunan Province Graduate Education Innovation Project and Professional Ability Enhancement Project Fund (Project Number: CX20200533); China Postdoctoral Science Foundation funded project（Project Number: 2017M622580). The funder had no role in study design, data collection and analysis, decision to publish, or preparation of the manuscript.

4.The data in this study are shared. We have generated supporting information files and will upload them together with the manuscript.

5.We have added a complete ethics statement in the Methods section of the manuscript.

6.Supporting information has been added to the end of the manuscript, and update any in-text citations to match accordingly.

Responds to the reviewer’s comments:

Reviewer #1:

Major:

Q1. Not enough attention has been paid to rigor in the control of genetic background. Comparisons are made between outcrossed animals and isogenic control animals without recognizing possible effects of hybrid vigor on phenotypes.

A1. We acknowledge that we did not use multiple GAL4 drivers to generate tissue-specific knockdowns. But before the study, we did some related work on the control of genetic background. In this study, we used the F1 generation crossed between W1118 and UAS-RNAi as a control. These flies are the offspring of a single cross. Secondly, we observed that the traits of the F1 generation of GAL4>UAS-RNAi are stable.

Q2. Rescue experiments are compared to each other (e.g. exercise to apoLpp RNAi) without including comparisons to wild type. This makes it impossible to quantitate the extent of rescue.

A2. In the rescue experiment, we supplemented the control group and quantified the degree of rescue. The specific content is in the revised manuscript.

Q3. No attention has been given to possible effects of any of the treatments on feeding rate.

A3. We carefully considered this issue and referred to the related article (DOI: 10.1371/journal.pone.0006063). The feeding rate of fruit flies is related to circadian rhythm, group size, and diet composition. In this experiment, we strictly follow the formula to make food for raising fruit flies. In addition, the fruit flies are reared in a constant incubator (25°C, 12 hours day and night). For the collected virgin flies, each bottle contained only 30 fruit flies to avoid the influence of group size on the feeding rate. In addition, in the exercise training intervention, fruit flies were allocated to spend 1.5 hours in a glass tube without food. At the same time, the fruit flies in the exercise-trained control group will also be allocated to a glass tube without food for 1.5 hours (they are in the same environment, the control group just has no exercise training). In short, the control of these conditions can overcome the influence of different treatments on the feeding rate of fruit flies to a certain extent.

Q4. Although experiments are done in females, no attention has been given to effects of reproduction on lipid accumulation. This is an essential point to address, because eggs are a substantial subset of the lipid accumulation in any female.

A4. At present, the study of heart-derived apoLpp for systemic lipid regulation is not comprehensive, and its mechanism is likely to be related to ovarian lipid synthesis. This is an excellent suggestion. Although the data of this experiment does not involve the effect of reproduction on lipid accumulation, we will further study the relationship between exercise-regulated heart-derived apoLpp and eggs in the future.

Q5. No rationale is given for limiting experiments to females.

A5. This is omitted in the introduction. There are two reasons why the subjects are restricted to females. Objective reason: In Drosophila, females have more triglyceride storage than males, and the triglyceride decomposition speed in response to lipolysis stimuli is slower. Subjective reasons: females are large and easy to observe and dissect. We have added relevant explanations in the Introduction.

Q6. Figures showing representative flies are clearly of different lengths and overall sizes, which surely must complicate assessment of whether they have "flatter abdomens". Abdominal bulging may result from reduced egg deposition and must be controlled for.

A6. We made some changes to this section. We added a chart of the wet weight of fruit flies on the basis of representative photos and used it to analyze the effects of different treatments on the bodyweight of fruit flies. The detailed revision is in the uploaded manuscript.

Q7. Oil Red O staining is not a quantitative assay for lipid content.

A7. We recognized this error and revised the description in the full text. We refer to a reasonable plan to use ORO to quantify abdominal lipids. (doi:10.1016/j.cub.2017.06.004.)

Q8. It is a major stretch to say that apoLpp RNAi phenotypes prove that effects of HFD are because of abnormal lipoprotein concentration.

A8. Indeed, this statement is wrong without the support of western blot experiments, and we have changed it in the article.

Q9. Discussion section does not effectively place the work in context, and basically restates the Introduction. Also, it is labelled "Discuss".

A9. For the discussion part, we reorganized the logic and revised it.

Minor: 

Q1. apoLpp should be spelled out and defined at first mention

A1. The spelling and definition of apoLpp have been added in the corresponding place.

Q2. Manuscript should be heavily edited for grammar.

A2. Due to the limitation of the native language, we handed over the manuscript to a professional polishing company for grammar revision.

Q3. Methods descriptions should be in past tense, and some of the statistical methods do not appear quite right. For example, pairwise comparisons after a 2-way ANOVA should use post-hoc adjustment, such as a Tukey test. All the comparisons in the paper appear to be pairwise, so it is not clear why the stats methods describe something different?

A3. This is a descriptive error. In the case of two factors, we used a two-way analysis of variance and used the LSD test for post-hoc adjustment.

Q4. What is interception length?

A4. We took a video of the heartbeat of fruit flies and output an electrocardiogram. "Interception length is 5s." This refers to the length of the electrocardiogram of 0-5s. During the re-production process, we increased the 5s to 10s and modified the corresponding description.

Q5. Does "half-exposure surgery" mean semi-intact preparation?

A5. Yes, we have changed to semi-intact preparation.

Q6. Description of Figure 1 refers incorrectly to figure 2 in several places, which is very confusing

A6. We have corrected this error and carefully compared the picture and text again.

Q7. Meaning of "superimposing effect" is not clear

A7. The superimposing effect means that regular exercise and apoLpp mRNA knockdown have a cumulative effect on the effect of systemic lipids. This statement was changed in the revised manuscript.

Reviewer #2:

Q1. The triglyceride (TG) assay method is not described correctly- there are no reagents listed and an antibody/substrate approach makes no sense in this protocol because triglycerides are not typical protein antigens. A standard curve should also be used in this assay so the ug TG/mg body weight can be reported. Without the proper methods, the data on TG cannot be evaluated.

A1. We re-described the determination method of triglycerides and added a standard curve in the supplementary information.

Q2. It would also be helpful if you described briefly how you isolated hearts; on line 210, it says that qPCR was used to quantify mRNA in cardiomyocytes, but the heart has nephrocytes and aliary muscles that are stuck to it and are sometimes included in the prep.

A2. We have added some details of separating the heart in the method section.

Q3. The heart rate of ~ 2 sec is surprising to me because most research papers have a rate of 2-3 beats per second. Perhaps the units should be Hz instead of seconds? Please double check this.

A3. I'm very sorry, this is a basic and serious mistake. The unit of heart rate is Hz. In addition, we checked the full text and corrected it.

Q4. For Fig 2B’s legend, “there is no accumulation of fat in the abdomen of flies in the HC-KD group” is not supported by the image. Reduced accumulation even looks like a stretch, but seems to be corroborated by the quantitative data in 2H provided that a legitimate TG assay was done.

A4. Indeed, ORO stained photos cannot directly quantify fat. We found a way to quantify the intensity of ORO staining using Photoshop (DOI: 10.1016/j.cub.2017.06.004). The specific modification is shown in the uploaded manuscript.

Q5. Figure 3’s title text “Regular exercise improves abnormal heart rhythm by reducing the expression of apoLpp mRNA in cardiomyocytes” is not supported by the data. The authors have only shown that apoLpp is reduced by exercise, not that this is the mechanism by which heart function is improved. For that, I think you’d have to overexpress apoLpp in the heart during exercise and show that heart function differs from the exercised control genotype. This experiment is one I would request if this were a different journal, but isn’t necessary to publish in PLoS One.

A5. We made some changes to the title of Figure 3 to make the description more accurate. “Regular exercise can reduce the expression level of apoLpp mRNA in cardiomyocytes and reverse the abnormal heart rhythm caused by the HFD diet”.

Q6. Fig 3G is TG and Fig 3H is apoLpp mRNA, but the legend has it the other way around.

A6. We have corrected this error.

Q7. With respect to the figure legends for panel A in Figs 1-4, I think it would be more clear if the text read simply “the abdomen is larger” (or smaller) rather than more prominent or flatter. It’s harder to think about how prominence and flatness are being measured and these might mean different things to different people. Large/small is straightforward- it would be even better if you graphed their wet or dry weights, data you might already have from the triglyceride assays. For 4A, remove significant; significant is reserved for statistical significance in my mind. You might say “no dramatic difference” instead. Again in 4A, I would focus on the size (smaller or larger), rather than “morphology,” which is not as easily ascertained from the images provided. There is only one view shown (perhaps the dorsal abdomen is spotted!) and the size is the most apparent difference in all of these panels.

A7. This is an excellent suggestion. We added a wet weight chart. This can be a good description of the information in Figure 1-4 panel A, and analyze the significance.

Q8. It would be of interest to directly compare control and apoLpp RNAi responses to exercise. This is difficult considering the way the figures are currently set up; perhaps it could be helpful to add a table or expand the Results or Discussion to mention these. I found the term “superimposed” difficult to understand. It could be rephrased as “Cardiac apoLpp is required for (some of) the benefits of exercise” – then you might consider why there is no longer a benefit from exercise. It may be because apoLpp RNAi hearts are more resilient during HF feeding. So cardiac apoLpp knockdown could be metabolically equivalent to exercising… this would be great, like an exercise mimetic. When I compare 3D and 4D, it looks like the heart rate is already an “exercised” value in the RNAi flies without exercise; the same seems to be true for fibrillation rates in 3F vs 4 F.

A8. We accept the reviewer's suggestion and delete the word "superimposed" and replace it with what is suggested in the comments. In addition, we also mentioned the response of apoLpp RNAi to exercise in the discussion section.

Q9. There are several small mistakes in writing that should be addressed- I will only mention a few here. Genes and mRNAs should be italicized. Line 110 should say reagent and line 111 and elsewhere should say H2O with a subscript #2. The use of female flies appears four times in the Fig 1 legend and three times in Figure 4’s legend and should be compressed to be as concise as possible. You can mention all flies are female early in the methods and results- then maybe in the Discussion (not Discuss, as on line 336) when comparing your findings to other high fat diet studies.

A9. We put "all fruit flies are female" in the method section. In addition, we also checked the full text and corrected minor flaws.

Q10. Finally, I am sleep deprived but I found it hard to keep it in my mind that HFD=HC. Why not use HFD throughout all of the text and figures? It would be easier for some people.

A10. We replaced HC in the full text with HFD.

---

## [Decision Letter · Decision Letter 1]

9 Nov 2021

PONE-D-21-21284R1Effects of Drosophila melanogaster  regular exercise and apolipoprotein B knockdown on abnormal heart rhythm induced by a high-fat dietPLOS ONE

Dear Dr.Zheng,

Thank you for submitting your manuscript to PLOS ONE. After careful consideration, we feel that it has merit but does not fully meet PLOS ONE’s publication criteria as it currently stands. Therefore, we invite you to submit a revised version of the manuscript that addresses the points raised during the review process. Both reviewers agreed that this is much improved version of the manuscript compared to the original submission. The authors did good job in responding my comments as well as questions raised by both reviewers. However, both reviewers still had some minor comments and need additional clarification. I agree with comments made by the reviewers. I also believe that addressing these minor comments should not take that long and authors should submit their revised version with in 30-days. 

We look forward to receiving your revised manuscript.

Kind regards,

Girish C. Melkani, Ph.D

Academic Editor

PLOS ONE

Journal Requirements:

Additional Editor Comments:

Both reviewers agreed that this is much improved version of the manuscript compared to the original submission. The authors did good job in responding my comments as well as questions raised by both reviewers. However, both reviewers still had some minor comments and need additional clarification. I agree with comments made by the reviewers. I also believe that addressing these minor comments should not take that long and authors should submit their revised version with in 30-days.

 **********

Review Comments to the Author

Reviewer #1: Authors have done an excellent job of addressing most of my concerns, I only have a few small comments remaining:

1. Authors have addressed my concerns about genetic background by clarifying that control flies are outcrossed RNAi lines, which is acceptable.

2. I am still a bit skeptical about the accuracy of Oil Red O intensity as a quantitation, and would recommend to authors that they only use this in cases where there is a clear and obvious qualitative difference in fat levels. However, since the authors have modified their methods to match those in published work, and since the results agree with the TAG assay, I think it's ok here.

3. I appreciate the addition of normal food groups in some figures to make it easier to see the degree of rescue.

4. My concern about feeding rate was not a concern about how the different groups were housed and fed, it was a concern about whether HFD or exercise themselves would alter feeding rate, which was not tested here, and would be a good control to add in future. Authors could at least reference other studies that have shown those controls, which may be good enough here.

The only change that I would insist on at this point is that the authors need to write their conclusions without stating that exercise acts through apoLpp to exert its effects on heart function and fat storage. All of these experiments are also consistent with these two things acting in parallel, and this is a key point. Just because exercise reduces apoLpp levels does not mean that this is the key mechanism of exercise effects. Authors should say something more like "apoLpp reduction may be involved in exercise-induced protection against HFD".

Reviewer #2: The authors have taken a number of steps to improve the manuscript, both with respect to the data and the writing. The data analysis and Discussion are significantly improved and I found most of the data compelling. The authors addressed most of my concerns and I still have significant concerns about the triglyceride assay, below. The other reviewer raised some very good points about the potential roles of feeding and egg laying that cannot be easily dismissed; however, these may fall outside the scope of the current manuscript.

Line 68 “sufficiently reduced to resist HFD-induced cardiac function impairment” is confusing. I think instead of resist they might mean elicit or potentiate.

line 87: w1118 has a lower case w.

Line 146. This TG assay is still of concern and my previous concern was not addressed. TG ELISAs seem to measure thyroglobulin, not triglycerides. There are not enough details to ascertain what was done. I do not believe triglycerides can be measured by an ELISA and the authors have given me no information to believe otherwise. A Google search for the kit listed produces nothing. What was used as the standard? Figs 1H and 2H should have a unit (ug TG/mg fly or something like this) not OD. OD isn’t as informative.

Line 164: How was Photoshop used to quantify the intensity of staining? Was a thresholding done to identify the red areas of the fat body cells? Were these dissociated from the cuticle before mounting and imaging? Some description would be helpful here.

Line 206: Figure 1: was this data generated in the control, transheterozygous genotype (UAS-Lpp RNAi x w1118 offspring)?

Line 288: The figure title is not fixed as the authors say in the response. The old, bad title is retained in the manuscript. Please use the new, improved title that you wrote in the reviewer response.

Line 298-300: the G and H descriptions need to be switched to match the figure.

Line 331- Figure 4: it doesn’t make sense to me to exclusively compare the normal diet control genotype to the HFD knockdown genotype, although I appreciate the fact that they are so close to each other. It seems as though the HFD control genotype should be included so the reader can see what knockdown and exercise does to each genotype. I realize this may be duplicative of the data shown in other panels, but if possible, it’s easiest to see differences when data are compared side-by-side.

Line 333: superimposed effects could read “combined effect”

Line 393: it reads “Knockdown of apoLpp in cardiomyocytes… does not directly affect the heart” I do not think this conclusion can be made given the data shown.

Line 400: reads “Flies allowed to exercise regularly show similar results to mammals with regard to high-fat- or high-sugar-induced heart damage.” The references provided do not seem to show these results.

The tracked changes would be easier for me to read without the old text in place; it’s easiest if the replacement text is highlighted.

---

## [Author Response · Author response to Decision Letter 1]

18 Nov 2021

List of Responses

Responds to the editor's comments:

1.We have re-checked the references in accordance with the requirements of the journal. We deleted a retracted paper and found a replacement.

Responds to the reviewer’s comments:

Reviewer #1:

Q1. My concern about feeding rate was not a concern about how the different groups were housed and fed, it was a concern about whether HFD or exercise themselves would alter feeding rate, which was not tested here, and would be a good control to add in future. Authors could at least reference other studies that have shown those controls, which may be good enough here.

A1. We have indeed neglected to control the feeding rate. At first, the establishment of the high-fat model was based on the research of Bird RT, 2010. In more than ten years of progress, the control of the feeding behavior of flies has become stricter, because the intake of the right type and amount of food has a major impact on the quality of life. In the future, we will focus on controlling the feeding rate of flies in order to obtain more convincing data. In addition, we explained this in the manuscript, and the specific content is highlighted in red.

Q2. The only change that I would insist on at this point is that the authors need to write their conclusions without stating that exercise acts through apoLpp to exert its effects on heart function and fat storage. All of these experiments are also consistent with these two things acting in parallel, and this is a key point. Just because exercise reduces apoLpp levels does not mean that this is the key mechanism of exercise effects. Authors should say something more like "apoLpp reduction may be involved in exercise-induced protection against HFD".

A2. The conclusion is indeed not concise and accurate enough. We very much agree with the reviewer's point of view and have made corresponding amendments in the conclusion.

Reviewer #2:

Q1. Line 68 “sufficiently reduced to resist HFD-induced cardiac function impairment” is confusing. I think instead of resist they might mean elicit or potentiate.

A1. I read the original text again and revised this sentence.

"Fibrillation of the Drosophila heart may result from lipotoxic damage related to the insulin-TOR signal, which is moderate reduction in insulin-TOR signaling prevents HFD-induced obesity and cardiac dysfunction"

Q2. Line 87: w1118 has a lower case w.

A2. We made a change to this and highlighted it in red in the text.

Q3. Line 146. This TG assay is still of concern and my previous concern was not addressed. TG ELISAs seem to measure thyroglobulin, not triglycerides. There are not enough details to ascertain what was done. I do not believe triglycerides can be measured by an ELISA and the authors have given me no information to believe otherwise. A Google search for the kit listed produces nothing. What was used as the standard? Figs 1H and 2H should have a unit (ug TG/mg fly or something like this) not OD. OD isn’t as informative.

A3. Thank you very much for your reminder. We realize that ELISA is not an effective way to quantify TG levels compared with the use of biochemical methods. We have added detailed experimental steps in Materials and Methods. In future research, we will circumvent this problem and choose more suitable experimental methods to quantify the data. In addition, we are willing to provide all the information on the kit for reviewers to view. The kit (Insect TG ELISA Kit) we use is produced by Shanghai Enzyme-linked Biotechnology, https://www.mlbio.cn/. We upload the kit instructions with the manuscript.

Q4. Line 164: How was Photoshop used to quantify the intensity of staining? Was a thresholding done to identify the red areas of the fat body cells? Were these dissociated from the cuticle before mounting and imaging? Some description would be helpful here.

A4. We added detailed methods in the manuscript. Invert the color image into a black/white image to identify the stained area, and use the average pixel density to quantify the staining intensity. In addition, before installation and imaging, the damage to adipose tissue should be reduced as much as possible to ensure the accuracy of the data. All changes are highlighted in red font in the materials and methods section.

Q5. Line 206: Figure 1: was this data generated in the control, transheterozygous genotype (UAS-Lpp RNAi x w1118 offspring)?

A5. Yes, we have rewritten the title of Figure 1 to make it more clear. “UAS-apoLpp RNAi>W1118 group was exposed to HFD for 5 days, resulting in obesity and abnormal heart rhythm”

Q6. Line 288: The figure title is not fixed as the authors say in the response. The old, bad title is retained in the manuscript. Please use the new, improved title that you wrote in the reviewer response.

A6. I personally express my apologies, this is a mistake. I have replaced the title of Figure 3 with the latest version.

Q7. Line 298-300: the G and H descriptions need to be switched to match the figure.

A7. The G and H in Figure 3 have been modified to their correct positions.

Q8. Line 331- Figure 4: it doesn’t make sense to me to exclusively compare the normal diet control genotype to the HFD knockdown genotype, although I appreciate the fact that they are so close to each other. It seems as though the HFD control genotype should be included so the reader can see what knockdown and exercise does to each genotype. I realize this may be duplicative of the data shown in other panels, but if possible, it’s easiest to see differences when data are compared side-by-side.

A8. In order to solve this problem, we reformatted Figure 4 and added the HFD control group on the original basis. As the reviewer said, this combination makes it easier to see the difference.

Q9. Line 333: superimposed effects could read “combined effect”

A9. I agree with your point of view and made corresponding changes in the manuscript.

Q10. Line 393: it reads “Knockdown of apoLpp in cardiomyocytes… does not directly affect the heart” I do not think this conclusion can be made given the data shown.

A10. This is a wrong conclusion, and we have deleted it from the manuscript. It should have been deleted in the first revision, but it was accidentally retained due to our negligence.

Q11. Line 400: reads “Flies allowed to exercise regularly show similar results to mammals with regard to high-fat- or high-sugar-induced heart damage.” The references provided do not seem to show these results.

A11. We re-read the cited article and found that this sentence is indeed an extension. We rewritten the sentences and added new quotes.

Q12. The tracked changes would be easier for me to read without the old text in place; it’s easiest if the replacement text is highlighted.

A12. We upload the manuscript, the manuscript with tracking changes, and the instructions for the ELISA kit to PLOS ONE.

---

## [Editor Report · Decision Letter 2]

26 Dec 2021

Effects of Drosophila melanogaster  regular exercise and apolipoprotein B knockdown on abnormal heart rhythm induced by a high-fat diet

PONE-D-21-21284R2

Dear Dr. Zheng,

We’re pleased to inform you that your manuscript has been judged scientifically suitable for publication and will be formally accepted for publication once it meets all outstanding technical requirements.

Kind regards,

Girish C. Melkani, Ph.D

Academic Editor

PLOS ONE
---

## [Editor Report · Acceptance letter]

4 May 2022

PONE-D-21-21284R2 

Effects of *Drosophila melanogaster* regular exercise and apolipoprotein B knockdown on abnormal heart rhythm induced by a high-fat diet 

Dear Dr. Zheng:

I'm pleased to inform you that your manuscript has been deemed suitable for publication in PLOS ONE. Congratulations! Your manuscript is now with our production department. 

Kind regards, 

on behalf of

Dr. Girish C. Melkani 

Academic Editor

PLOS ONE